# Knowledge, attitude and practice of hepatitis B infection prevention among nursing students in the Upper West Region of Ghana: A cross-sectional study

**Augustine Ngmenemandel Balegha[1]\*, Adadow Yidana[2‡], Gilbert Abotisem Abiiro[3‡]**

**1** Department of Obstetrics and Gynaecology, Upper West Regional Hospital, Wa, Upper West Region, Ghana, **2** Department of Behavioural and Social Change, School of Public Health, University for Development Studies, Tamale, Northern Region, Ghana, **3** Department of Health Services, Policy, Planning, Management and Economics, School of Public Health, University for Development Studies, Tamale, Northern Region, Ghana

‡ These authors also contributed equally to this work.
\* bnaugustine@gmail.com

**Data Availability Statement:** All relevant data are within the manuscript and its Supporting Information files.

## Abstract

### Introduction

Hepatitis B infection remains a public health threat associated with undesirable statistics of morbidity and mortality. Good knowledge, attitude and practice (KAP) of hepatitis B infection (HBI) prevention are essential for HBI control. However, there is limited evidence concerning the KAP of HBI prevention among nursing students, who are significantly exposed to HBI. We assessed the KAP of HBI prevention and the factors associated with the practice of HBI prevention among nursing students in the Upper West Region of Ghana.

### Methods

We administered an online cross-sectional survey in November 2020 to a stratified random sample of 402 nursing students in two nursing training colleges in the Upper West Region. Using STATA version 13, we computed composite scores of KAP of HBI prevention with maximum scores of 18 for knowledge and 8 each for attitude and practice. A generalised ordered logistic regression model was run to assess the factors associated with the practice of HBI prevention.

### Results

The students had moderate median scores for knowledge (12.00; IQR = 10–13) and attitude (6.00; IQR = 5.00–7.00) but a poor median score (5.00; IQR = 4.00–6.00) for the practice of HBI prevention. High knowledge (aOR = 2.05; $p$ = 0.06), good attitude, being a male, second year student and having parents with tertiary education were significantly associated with higher likelihoods (aOR >1; $p$ < 0.05) of demonstrating good practice of HBI prevention. Students who had never married were significantly (aOR = 0.34; $p$ = 0.010) less likely to exhibit good practice of HBI prevention.

**Funding:** The authors received no specific funding for this work.

**Competing interests:** The authors have declared that no competing interests exist.

## Conclusion

The KAP scores of HBI prevention among the students were sub-optimal. We recommend institution-based policies and regular education on HBI prevention, free/subsidised HBI prevention services, and the enforcement of proper professional ethics on HBI prevention in nursing training colleges. Such interventions should predominantly target female, non-married and first year nursing students.

## Introduction

Hepatitis B infection is a contagious disease of the liver, caused by the partial double-stranded deoxyribonucleic acid (dsDNA) hepatitis B virus (HBV) and transmitted from an infected mother to her child at birth, through needle stick injuries (NSI), unsterilised surgical instruments, exposure to infected blood and bodily fluids, and sexual intercourse [1]. Viral hepatitis including hepatitis B infection has been targeted for elimination by 2030, as provided by Sustainable Development Goal three and the Global Health Sector Strategy on viral hepatitis 2016–2021 [2,3].Globally, hepatitis B infection affects about a third of the world's population and causes 1.34 million mortalities annually, being the 7th leading cause of mortality worldwide [4]. It is estimated that about 250 million persons are chronically infected with hepatitis B virus worldwide [1]. Chronic hepatitis B infection increases the risk of developing complications such as liver cirrhosis and hepatocellular cancer which have been long established as the major contributors to hepatitis B infection specific mortalities [5]. In sub-Saharan Africa, the prevalence of hepatitis B infection is estimated to be 6.1% with an associated 87,890 mortalities annually [4]. In Ghana, several studies have reported a higher hepatitis B infection prevalence well over 10% among different study groups including healthcare workers [6,7].

Effective hepatitis B infection prevention is a product of high knowledge, good attitude and good practice of hepatitis B infection prevention [8]. Knowledge of hepatitis B infection prevention in this context is the awareness and understanding of the aetiology of hepatitis B infection, its transmission, clinical manifestation, diagnosis, treatment, complications, vaccination and existence of post exposure prophylaxis for the management of accidentally exposed persons [5]. Attitude towards hepatitis B infection prevention in this context is predicated on perceived susceptibility/risk, perceived severity and perceived threat of hepatitis B infection [5,9]. Good practice of hepatitis B infection prevention is the uptake of hepatitis B infection prevention activities such as hepatitis B screening, hepatitis B vaccination, post-hepatitis B vaccination antibody testing, changing of gloves per client, non-recapping of needles after use, prevention of NSI, and prevention of blood splashes on body.

High knowledge and good attitude towards disease prevention moderated by relevant socio-demographics, following health education have been demonstrated to be positively related to good practice of disease prevention, which is hypothesised to culminate in desirable disease prevention outcome [8]. These relationships have been largely drawn from the knowledge attitude practice-outcome (KAP-O) framework proposed by Wan [10] and furthered by Rav-Marathe et al., [8].

A review of empirical literature revealed that globally, there is a paucity of studies on KAP of hepatitis B infection prevention, particularly in Ghana [11,12], and of the factors associated with the practice of hepatitis B infection prevention [13,14]. Nonetheless, these few studies have largely been conducted among healthcare workers s and not trainees of the healthcare profession including nursing students [11,15]. Although healthcare workers are known to be

four times at risk of hepatitis B infection compared to the general population [16], their trainees including nursing students have been acknowledged to carry a greater risk of infection by HBV, due to inexperience, inadequate training and sheer carelessness [17,18]. This study, therefore, assessed the KAP of hepatitis B infection prevention and the factors associated with the practice of hepatitis B infection prevention among nursing students in the Upper West Region of Ghana.

## Methods

### KAP-O framework of the study

According to the KAP-O framework proposed by Wan [10] and advanced by Rav-Marathe et al. [8], an interactional relationship exists among knowledge, attitude, practice and outcomes of disease prevention. The KAP-O framework postulates that, educational interventions such as health education can improve knowledge of and attitude towards disease prevention, which can translate into good practice of disease prevention and ultimately lead to improved outcomes of disease prevention.

Our adapted KAP-O framework (Fig 1) illustrates that nursing students' knowledge (of the aetiology of hepatitis B infection, transmission, clinical manifestation, diagnosis, treatment, complications, vaccination and post exposure prophylaxis for hepatitis B infection) and attitudes (perceived susceptibility, severity and threat) towards hepatitis B infection prevention, interact with their socio-demographic characteristics including health education, to influence their practices (hepatitis B screening, vaccination, vaccination dosage, post-vaccination testing, changes of gloves per client, recapping of needles and prevention of NSI) of hepatitis B infection prevention. This interaction ultimately culminates in desirable hepatitis B infection prevention outcomes which is beyond the scope of this study. Guided by existing literature [12,13,19], the socio-demographic characteristics considered in the study include sex, age, marital status, religion, ethnicity, residential setting, college of enrolment, programme of

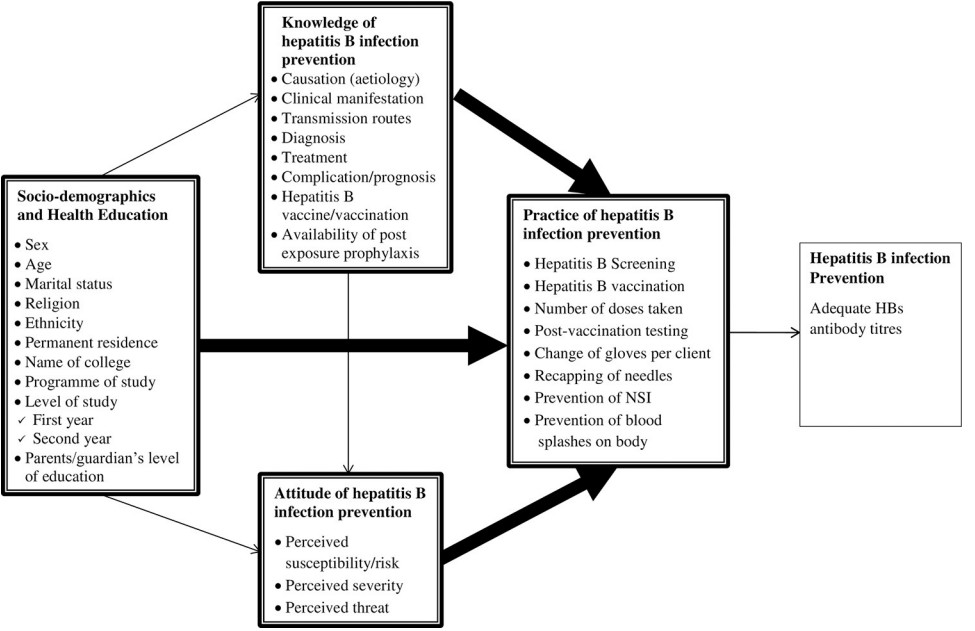

**Fig 1. Conceptual framework of the study.**

study, level of study and parents' level of education. Although not covered in this study, our conceptual framework also recognises that knowledge of hepatitis B infection prevention can influence attitudes towards hepatitis B infection prevention.

## Study setting, approach and design

This study was conducted in Ghana, specifically in the Wa and Lawra Municipalities of the Upper West Religion. The Upper West Region has seven state owned health training institutions, located in four of eleven districts of the region. This study was conducted among nursing students of Wa Nursing Training College (NTC) located in the Wa Municipality, and Lawra NTC located in the Lawra Municipality. Wa and Lawra NTCs were purposively selected based on the type of setting and programmes of study. Wa NTC is located in a predominantly urban district while Lawra NTC is located in a predominantly rural district. At the time of data collection, Wa NTC had a student population of 399 and Lawra NTC had a student population of 303. Both colleges train two cadres of nurses via their three-year Registered General Nursing (RGN) and two-year Nurse Assistant Curative (NAC) programmes. However, the final year students of both colleges were excluded from the study due to their unavailability since they had started their final examination. Our study used the quantitative approach and an analytical cross-sectional design.

## Sampling

The sample size for this study was determined using Cochran [20]'s formula;

$$n = z^2 pq/d^2$$

where n is the sample size, $z = 1.96$, p = Estimated proportion of hepatitis B vaccination coverage among the nursing students = 66.8% [17] q = 1-p and d = margin of error of 5%. Therefore, the sample size for the study was 341. Assuming a 20% non-response rate, the sample size was estimated to be 410. The respondents were recruited using a multistage stratified random sampling technique. The study population was stratified based on the name of college, the programme of study and level of study. Proportionate sample size was calculated for the students from each stratum. A simple random sampling of the students in each stratum was done using an index number-based class register. This was done with a random table of numbers until the required sample size was obtained.

## Data collection

The data collection instrument was a self-administered structured questionnaire (**S1 Questionnaire**) with closed-ended questions. The questionnaire was adapted from previous studies [5,21,22]. It consisted of 50 questions. Section A assessed the socio-demographic characteristics of the respondents with 10 questions, covering sex, age, marital status, religion, ethnicity, permanent residential setting, name of college, programme of study, level of study and parent's level of education. Section B covered eight variables on the level of knowledge of the respondents on hepatitis B infection prevention using 18 questions which elicited "yes, no and not sure" responses. The eight variables were causation (aetiology), clinical manifestation, transmission routes, diagnosis, treatment, complications/prognosis, hepatitis B vaccine/vaccination and post-exposure prophylaxis. Section C covered three variables (perceived susceptibility, perceived severity and perceived threat) on the attitude of the nursing students towards hepatitis B infection prevention, using eight questions which elicited five-point Likert scale responses of "strongly agree, agree, uncertain, disagree and strongly disagree". Section D consisted of

eight questions on the practice of hepatitis B infection prevention, covering eight variables including hepatitis B screening, hepatitis B vaccination, number of doses, post vaccination testing, changing of gloves, recapping of needles, acquisition of NSI and blood splashes on the body.

Due to the prevailing COVID-19 pandemic, the study was mostly an online survey but respondents who lacked data for internet connectivity were supplied hard copies of the questionnaire to complete. WhatsApp contacts of the randomly sampled respondents were obtained in consultation with their course tutors. On the day of data collection, informed consent was obtained and the questionnaire on google forms was sent to each respondent via WhatsApp. Non-responders were reminded on weekly basis and encouraged to voluntarily complete the questionnaire [23]. The data collection exercise lasted for three weeks, from 6th to 27th November, 2020.

## Validity and reliability of the instrument

The instrument was adapted from several studies after a thorough literature review [5,21,22]. The authors examined the questionnaire for face and content validity and ensured that the questionnaire appropriately addressed all relevant questions pertaining to the study within the local context. The instrument was pretested among 25 nursing students of Jirapa Nursing Training College (also in the Upper West Region of Ghana), who had similar socio-demographic characteristics as the actual study participants. **S1 Data** contains the dataset from the pre-test. Feedback from the pre-test assisted in recasting unintelligible statements/questions, correction of grammatical errors and appreciation of the length of time required to complete the actual survey. Construct validity was assessed using inter-item correlation [24]. The average total correlation co-efficient of the items for knowledge, attitude and practice of hepatitis B infection prevention were $r_k$ = 0.440, $r_a$ = 0.528 and $r_p$ = 0.560 respectively, which were considered acceptable [24]. Internal consistency (reliability) of the measurement scales of the instrument was computed using Cronbach [25]'s alpha ($\alpha$). The overall alpha of the instrument was $\alpha$ = 0.71 while the alpha values for the scales of knowledge, attitude and practice of hepatitis B infection prevention were $\alpha$ = 0.77, $\alpha$ = 0.66 and $\alpha$ = 0.70 respectively.

## Ethical considerations

The study was granted ethical approval by the Committee on Human Research Publications and Ethics of the Kwame Nkrumah University of Science and Technology/School of Medical Sciences (CHRPE/AP/392/20). Institutional access approval was also obtained from the Upper West Regional Health Directorate as well as the Principals of Wa and Lawra Nursing Training Colleges. Written informed consent was obtained from all respondents. Our survey was conducted in line with the principles of the Declarations of Helsinki.

## Data analyses

The responses to the survey were recorded digitally and displayed as spreadsheets. The data was cleaned, coded and imported to STATA version 13 computer software for analysis.

Descriptive analysis of the socio-demographic characteristics of the respondents and the KAP of hepatitis B infection prevention was performed using frequencies and percentages. Composite scores for KAP of hepatitis B infection prevention were computed by summing up the scores obtained by each respondent. A correctly answered question was awarded 1 point while a wrongly answered question was awarded no point (**S1 Table**). The minimum obtainable composite score for knowledge was 0 while the maximum obtainable composite score was 18. For attitude and practice of hepatitis B infection prevention, the minimum and maximum

obtainable composite scores were 0 and 8 respectively. Summary statistics of mean (standard deviation [SD]), median (interquartile range [IQR]), minimum, maximum, range, skewness and kurtosis were computed for the scores of KAP of hepatitis B infection prevention. However, considering that the composite scores of the KAP of hepatitis B infection prevention were non-normally distributed, median composite scores were used as measures of central tendencies [26].

In line with Osei et al. [27], the composite scores for knowledge, attitude and practice of hepatitis B infection prevention were each re-categorised into three. Knowledge of hepatitis B infection prevention was categorised as low, moderate and high, corresponding with composite scores of 0–9, 10–14, 15–18 respectively. Scores for attitude and practice were each categorised as poor, moderate and good to correspond with composite scores of 0–4, 5–6 and 7–8 respectively.

A generalised ordered logistic regression model was run to determine the factors associated with the practice of hepatitis B infection prevention. The model assessed the associations of the independent variables (the categorised scores of knowledge and attitude of hepatitis B infection prevention, and the socio-demographic characteristics of the respondents) with the practice (categorised scores) of hepatitis B infection prevention. Since, the practice of hepatitis B infection prevention, the dependent variable in this study, was categorised as a polychotomous ordered categorical (poor, moderate and good practice) response variable, a multivariate ordered logistic regression (OLOGIT) model was run. The OLOGIT model with log pseudo-likelihood of -397.00563 was statistically significant; $\chi^2$ (15) = 62.01, $p < 0.0001$ and explained 8.23% (McFadden $R^2$) of the variance in the practice of hepatitis B infection prevention. However, for the results of an OLOGIT model to be valid, it must satisfy the parallel regression/proportional odds assumption by passing the Brant test [28]. The parallel regression/proportional odds assumption assumes that the model slopes/estimates (β-coefficients/odds ratios) generated across the different ordinal outcome categories (for example poor and moderate combined versus good practice, and poor versus moderate and good practice combined) are constant while the intercepts are variable [28]. The Brant test essentially assesses the global significance of the model as well as the individual significance of all the explanatory variables in the model, with a significant test results implying invalidity [29]. Our OLOGIT model failed the Brant test of proportional odds assumption by producing a global statistically significant test statistic (*p = 0.017*). Therefore, a default GOLOGIT2 model was run [29]. The GOLOGIT2 model is an unconstrained model in which the intercepts and slopes (β- coefficients/odds ratios) show variations across the ordinal outcome categories [29]. The GOLOGIT2 model we ran, although better fitting with superior explanatory power when compared with the OLO-GIT model (Log pseudo-likelihood = -374.19736; McFadden's pseudo $R^2$ = 0.1350), was limited in the context of yielding cumbersome extraneous estimates [30]. Therefore, a GOLOGIT2 model with autofit option was run as a preferred alternative [30]. Similar to Brant test, Wald test of partial parallel regression/partial proportional odds assumption, must be satisfied by the GOLOGIT2 model in order for the model to be valid [29]. The GOLOGIT2/autofit model essentially put constraints on the variables that met Wald test of partial parallel regression/partial proportional odds assumption, while relaxing those that violated the assumption [29]. In our GOLOGIT2/autofit model, three variables out of fifteen variables (NAC programme of study, first year of study and below tertiary parent education status) failed the Wald test by producing significant test statistics (*p < 0.05*) and hence constraints for parallel lines were not imposed (relaxed) on these three variables by the autofit function of the GOLOGIT2 model. Therefore, the two outputs of the GOLOGIT2 autofit model, are depicted in the results section as GOLOGIT2/autofit1 (initial output) and GOLOGIT2/autofit2 (final and interpretable output). Although less restrictive, the GOLOGIT2/autofit model is more

parsimonious than a multinomial logit model [30]. The regression results are presented in Adjusted Odds Ratios (aORs) and statistical significance determined at 95% confidence interval (CI).

## Results

### Socio-demographic characteristics of the nursing students

Of the 410 nursing students sampled, 402 responded to the survey. This represents a response rate of 98.05%. The non-responses were due to refusals to participate in the survey with the reason of not being interested. Table 1 presents the socio-demographic characteristics of the respondents. Majority of the respondents (71.4%) were females. 54.5% of them were aged $\geq$ the median age of 23 years. 90.8% of them had never been married. Majority (71.4%) of them were first year students. Only 41.3% of the parents/guardians of the students had attained tertiary education status.

### Knowledge, attitude and practice of hepatitis B infection prevention

Table 2 presents the nursing students' knowledge of hepatitis B infection prevention. Aetiologically, only 41.0% of the nursing students knew that hepatitis B infection is caused by a dsDNA virus. The respondents demonstrated varied knowledge of the following clinical manifestations of hepatitis B infection: infectiousness of hepatitis B infection carriers (75.9%), jaundice as a symptom of hepatitis B infection (59.7%) and not all infected persons become symptomatic (58.7%). In terms of transmission, the nursing students recognised contaminated blood/products (90.8%), unsterilised surgical instruments (85.6%), unprotected sex (81.1%) and mother-to-child transmission (79.8%) as the major routes of transmission of hepatitis B

**Table 1. Socio-demographic characteristics of the nursing students (n = 402).**

| Variables | Categories | Frequency (%) |
|---|---|---|
| Sex | Female | 287 (71.4) |
|  | Male | 115 (28.6) |
| Age | < 23 years | 183 (45.5) |
|  | $\geq$ 23 years | 219 (54.5) |
| Marital status | Married | 37 (9.2) |
|  | Never married | 365 (90.8) |
| Religion | Islam | 153 (38.1) |
|  | Christianity | 249 (61.9) |
| Ethnicity | Dagaaba/Waala/Sisaala | 223 (55.5) |
|  | Akan | 98 (24.4) |
|  | Others | 81 (20.1) |
| Residential setting | Rural | 194 (48.3) |
|  | Urban | 208 (51.7) |
| Name of College | Lawra NTC | 177 (44.0) |
|  | Wa NTC | 225 (56.0) |
| Programme of Study | NAC | 201 (50.0) |
|  | RGN | 201 (50.0) |
| Level of study | First year | 287 (71.4) |
|  | Second year | 115 (28.6) |
| Parents' level of education | Below Tertiary | 236 (58.7) |
|  | Tertiary | 166 (41.3) |

**Table 2. Knowledge of hepatitis B infection prevention (n = 402).**

| Variable | Frequency (%) | |
|---|---|---|
| | **Yes** | **No** |
| **Causation** | | |
| Hep. B caused by dsDNA virus | 165 (41.0) | 237 (59.0) |
| **Clinical manifestation** | | |
| Jaundice as a symptom | 240 (59.7) | 162 (40.3) |
| Not all the infected are symptomatic | 236 (58.7) | 166 (41.3) |
| Infectiousness of carriers | 305 (75.9) | 97 (24.1) |
| **Transmission routes** | | |
| Not transmitted faeco-orally | 175 (43.5) | 227 (56.5) |
| Not transmitted by casual contact | 242 (60.2) | 160 (39.8) |
| Contaminated blood/products | 365 (90.8) | 37 (9.2) |
| Unsterilised surgical instruments | 344 (85.6) | 58 (14.2) |
| Unprotected sex | 326 (81.1) | 76 (18.9) |
| Mother-to-child transmission | 321 (79.8) | 81 (20.1) |
| **Diagnosis by** | | |
| Serology | 213 (53.0) | 189 (47.0) |
| PCR | 127 (31.6) | 275 (68.4) |
| **Management/Treatment** | | |
| Incurable | 249 (61.9) | 153 (38.1) |
| **Prognosis/complications** | | |
| Liver cancer | 308 (76.6) | 94 (23.4) |
| **Hepatitis B vaccine/Vaccination** | | |
| Not made from human blood | 113 (28.1) | 289 (71.9) |
| Prevents hepatitis B infection | 358 (89.1) | 44 (10.9) |
| Protects against liver cancer | 195 (48.5) | 207 (51.5) |
| **Post exposure prophylaxis** | | |
| Availability | 217 (54.0) | 185 (46.0) |

Yes = Correct response; No = Wrong response.

infection. Also, 60.2% and 43.5% of the respondents knew that casual contact and the faeco-oral route are not viable transmission routes of hepatitis B infection, respectively. Majority (53.0%) of the respondents knew the serological diagnostic method while only 31.6% of them knew the Polymerase Chain Reaction (PCR) diagnostic method. In terms of management, 76.6% of the students knew liver cancer as a complication of hepatitis B infection and 61.9% of them knew that hepatitis B infection is incurable. In terms of hepatitis B vaccination, only 28% of the respondents knew that hepatitis B vaccine is not derived from human blood. Majority (89.1%) of the students knew that hepatitis B vaccine prevents hepatitis B infection. However, only 48.5% of them knew that hepatitis B vaccine protects against liver cancer and 54.0% of the respondents knew that post-exposure prophylaxis exists for accidental exposure to hepatitis B infection.

Table 3 displays the attitudes of the nursing students towards hepatitis B infection prevention. In terms of perceived susceptibility, 57.5% of the respondents perceived themselves to be generally at risk of hepatitis B infection while 56.2% of them recognised that any exposure to blood is risky. Majority (87.8%) of them acknowledged that personal protective equipment usage is necessary during surgery. Regarding perceived severity/seriousness, majority (87.6%) of the respondents acknowledged that hepatitis B infection is more serious than HIV infection.

**Table 3.  Attitude towards hepatitis B infection prevention (n = 402).**

| Variables | Frequency (%) | |
|---|---|---|
| | **Yes** | **No** |
| **Perceived susceptibility** | | |
| Generally, you are at risk of infection | 231 (57.5) | 171 (42.5) |
| Occasional contact with blood is risky | 226 (56.2) | 176 (43.8) |
| Personal protective equipment usage is necessary in surgery | 353 (87.8) | 49 (12.2) |
| **Perceived Severity** | | |
| Hepatitis B infection more serious than HIV/AIDS | 352 (87.6) | 50 (12.4) |
| Hep B is serious in terms of living normal lives | 285 (70.9) | 117 (29.1) |
| Hepatitis B is serious regardless of treatability | 312 (77.6) | 90 (22.4) |
| **Perceived Threat** | | |
| Needle pricks require reporting | 300 (74.7) | 102 (25.4) |
| Blood splashes on body require reporting | 333 (82.8) | 69 (17.2) |

Agree = Correct response; Disagree = Wrong response.

Also, 70.9% perceived aptly that hepatitis B infection is serious in terms of living normal life. Majority (77.6%) of the respondents agreed that hepatitis B infection is serious regardless of its treatability. In respect of perceived threat, majority of the respondents agreed that blood splashes (82.8%) and NSI (74.7%) should be reported.

Table 4 shows the nursing students' practices or uptake of hepatitis B infection prevention activities. The prevalence of the practice of various hepatitis B infection prevention activities among the nursing students were reported as; hepatitis B screening (89.1%), changed gloves per client (84.1%), hepatitis B vaccination (72.1%), never splashed blood on body (72.1%), never acquired NSI (70.9%), full dose vaccination (59.5%), never recapped needles (21.4%) and post-hepatitis B vaccination antibody testing (19.4%).

Table 5 presents the summary statistics of the composite scores of overall KAP of hepatitis B infection prevention. The overall median score (with IQR) for level of knowledge of the respondents was moderate, (12.00; IQR = 10.00–13.00) out of a maximum of 18.00. The overall median attitude score of the nursing students was also moderate, (6.00; IQR = 5.00–7.00) out of a maximum of 8.00. The overall median practice score of hepatitis B infection prevention was poor, (5.00; IQR = 4.00–6.00) out of a maximum of 8.00. KAP of hepatitis B infection prevention scores showed negative skewness and leptokurtosis (positive kurtosis).

## Factors associated with the practice of hepatitis B infection prevention

Table 6 presents the results of the generalised ordered logit model on the factors associated with the practice of hepatitis B infection prevention. As shown in Table 6, high knowledge of ($p$ = 0.060) and good attitude towards hepatitis B infection prevention, male sex, never married status, second year in nursing college and tertiary education status of parents ($p < 0.05$) were statistically significantly associated with the practice of hepatitis B infection prevention, provided all other variables in the model remain constant. Compared to those nursing students with low level of knowledge of hepatitis B infection prevention, those with high level of knowledge were about two times more likely to perform good practices of hepatitis B infection prevention versus that of moderate and poor practices combined (aOR = 2.05; 95% CI = 0.97–4.33; $p$ = 0.060). Compared to those nursing students with poor attitudes towards hepatitis B infection prevention, those with good attitudes were about three times more likely to exhibit good practices of hepatitis B infection prevention versus that of moderate and poor practices

**Table 4. Practice of hepatitis B infection prevention (n = 402).**

| Variables | Categories | Frequency (%) |
|---|---|---|
| Hepatitis B screening | Yes | 358 (89.1) |
| | No | 44 (10.9) |
| Hepatitis B vaccination | Yes | 290 (72.1) |
| | No | 112 (27.9) |
| Number of doses | Zero | 112 (27.9) |
| | One dose | 29 (7.2) |
| | Two doses | 22 (5.5) |
| | Three doses | 234 (58.2) |
| | Four doses | 5 (1.2) |
| Post vaccination testing | Yes | 78 (19.4) |
| | No | 212 (52.7) |
| | Not Applicable | 112 (27.9) |
| Changes gloves | Always | 338 (84.1) |
| | Sometimes | 47 (11.7) |
| | Never | 17 (4.2) |
| Recapping of needles | Always | 250 (62.2) |
| | Sometimes | 66 (16.4) |
| | Never | 86 (21.4) |
| Acquisition of NSI | Always | 18 (4.5) |
| | Sometimes | 99 (24.6) |
| | Never | 285 (70.9) |
| Blood splashes on body | Always | 27 (6.7) |
| | Sometimes | 85 (21.1) |
| | Never | 290 (72.1) |

combined (aOR = 2.52; 95% CI = 1.44–4.41; $p$ = 0.001). Compared to female nursing students, male nursing students were about two times more likely to exhibit good practices of hepatitis B infection prevention versus that of moderate and poor practices combined (aOR = 1.92; 95% CI = 1.23–3.03; $p$ = 0.004). Also, compared to first-year nursing students, second-year nursing students were about three times more likely to perform good practices of hepatitis B infection prevention versus that of moderate and poor practices combined (aOR = 2.68; 95% CI = 1.19–6.03; $p$ = 0.017). Furthermore, compared to those whose parents had below tertiary education, those students whose parents had tertiary education were about four times more likely to perform good practices of hepatitis B infection prevention versus that of moderate and poor practices combined (aOR = 3.79; 95% CI = 2.26–6.37; $p$ < 0.0001). However, compared to married nursing students, those that were not married were less likely to exhibit good practices of hepatitis B infection prevention versus that of moderate and poor practices combined (aOR = 0.34; 95% CI = 0.15–0.77; $p$ = 0.010).

## Discussion

The overall level of knowledge of the respondents on hepatitis B infection prevention was moderate. This finding is similar to those of Adam and Fuseini [31] and Osei et al. [27] in Ghana. Hang Pham et al. [32] in China even reported poor knowledge of hepatitis B infection prevention among healthcare workers. This implies that hepatitis B infection prevention education obtained by these students was inadequate. This inadequacy in respect of hepatitis B infection prevention education plausibly did not provide the enabling environment and

**Table 5. Composite scores of overall KAP of hepatitis B infection prevention (n = 402).**

| Variables | Frequency (%) |
|---|---|
| **Knowledge of hepatitis B infection prevention** | |
| Low | 102 (25.3) |
| Moderate | 264 (65.7) |
| High | 36 (9.0) |
| Mean (SD) | 11.23 (± 2.88) |
| Median (IQR) | 12.00 (10.00–13.00) |
| Minimum | 3.00 (0.8%) |
| Maximum | 17.00 (2.4%) |
| Range | 14 (17–3.00) |
| Skewness | -0.494 |
| Kurtosis | 2.960 |
| **Attitude towards hepatitis B infection prevention** | |
| Poor | 83 (20.6) |
| Moderate | 196 (48.8) |
| Good | 123 (30.6) |
| Mean (SD) | 5.71 (± 1.53) |
| Median (IQR) | 6.00 (5.00–7.00) |
| Minimum | 0.00 (0.8%) |
| Maximum | 8.00 (11.2%) |
| Range | 8.00 (8.00–0.00) |
| Skewness | -0.750 |
| Kurtosis | 3.873 |
| **Practice of hepatitis B infection Prevention** | |
| Poor | 168 (41.8) |
| Moderate | 135(33.6) |
| Good | 99 (24.6) |
| Mean (SD) | 4.90 (± 1.77) |
| Median (IQR) | 5.00 (4.00–6.00) |
| Minimum | 0.00 (0.5%) |
| Maximum | 8.00 (1.5%) |
| Range | 8.00 (8.00–0.00) |
| Skewness | -0.351 |
| Kurtosis | 2.218 |

capacity needed by the nursing students to exercise control over their own health. Poor control over their health arising from low knowledge is non-optimal for effective action towards disease prevention. Therefore, regular education on hepatitis B infection prevention, which has the desirable implication of promoting disease prevention, is crucial to the empowerment of these nursing students.

Our study also reported an overall moderate level of attitude of the respondents towards hepatitis B infection prevention. This finding of ours is similar to that of Akazong et al. [33] in Cameroun. This moderate perception of the risk, severity and threat of hepatitis B infection prevention by the nursing students implies that such students maybe unwilling to develop personal skills relevant to hepatitis B disease prevention, even in the presence of a healthy hepatitis B infection prevention policy and a supportive environment, created through education and subsidised preventive healthcare services. Almualm et al. [34]'s study even reported a poor overall attitude towards hepatitis B infection prevention among Yemeni medical students.

**Table 6. Factors associated with the practice of hepatitis B infection prevention (n = 402).**

| Variables | GOLOGIT2auto1 | | | GOLOGIT2auto2 | | |
|---|---|---|---|---|---|---|
| | aOR | 95% CI | p-value | aOR | 95% CI | p-value |
| **Knowledge** | | | | | | |
| Low[1] | 1 | | | 1 | | |
| Moderate | 0.88 | 0.55–1.40 | 0.590 | 0.88 | 0.55–1.40 | 0.590 |
| High | 2.05 | 0.97–4.33 | **0.060** | 2.05 | 0.97–4.33 | **0.060** |
| **Attitude** | | | | | | |
| Poor[1] | 1 | | | 1 | | |
| Moderate | 0.82 | 0.49–1.37 | 0.440 | 0.82 | 0.49–1.37 | 0.440 |
| Good | 2.52 | 1.44–4.41 | **0.001** | 2.52 | 1.44–4.41 | **0.001** |
| **Sex** | | | | | | |
| Female[1] | 1 | | | 1 | | |
| Male | 1.93 | 1.23–3.03 | **0.004** | 1.93 | 1.23–3.03 | **0.004** |
| **Age** | | | | | | |
| < Median (23)[1] | 1 | | | 1 | | |
| ≥ Median | 0.81 | 0.54–1.23 | 0.328 | 0.81 | 0.54–1.23 | 0.328 |
| **Marital status** | | | | | | |
| Married[1] | 1 | | | 1 | | |
| Never married | 0.34 | 0.15–0.77 | **0.010** | 0.34 | 0.15–0.77 | **0.010** |
| **Religion** | | | | | | |
| Islam[1] | 1 | | | 1 | | |
| Christianity | 1.15 | 0.70–1.89 | 0.580 | 1.15 | 0.70–1.89 | 1.580 |
| **Ethnicity** | | | | | | |
| Dagaaba/Waala/Sisaala[1] | 1 | | | 1 | | |
| Akan | 1.45 | 0.88–2.39 | 0.140 | 1.45 | 0.88–2.39 | 0.140 |
| Others | 1.15 | 0.71–1.88 | 0.567 | 1.15 | 0.71–1.88 | 0.567 |
| **Residence** | | | | | | |
| Rural[1] | 1 | | | 1 | | |
| Urban | 0.87 | 0.58–1.30 | 0.500 | 0.87 | 0.58–1.30 | 0.500 |
| **College** | | | | | | |
| Lawra NTC[1] | 1 | | | 1 | | |
| Wa NTC | 1.20 | 0.77–1.84 | 0.420 | 1.20 | 0.77–1.84 | 0.420 |
| **Programme** | | | | | | |
| NAC[1] | 1 | | | 1 | | |
| RGN | 1.45 | 0.82–2.54 | 0.199 | 0.56 | 0.27–1.16 | 0.121 |
| **Level of study** | | | | | | |
| First year[1] | 1 | | | 1 | | |
| Second year | 0.92 | 0.48–1.75 | 0.799 | 2.68 | 1.19–6.03 | **0.017** |
| **Parent's education** | | | | | | |
| Below Tertiary[1] | 1 | | | 1 | | |
| Tertiary | 1.20 | 0.77–1.88 | 0.418 | 3.79 | 2.26–6.37 | **< 0.0001** |
| **Constant** | 1.94 | 0.61–6.22 | 0.261 | 0.24 | 0.74–0.79 | **0.019** |
| **Observations** | | 402 | | | | |
| **Wald chi² (18)** | | 78.49 | | | | |
| **Prob > chi²** | | 0.0000 | | | | |
| **Log pseudo-likelihood** | | -381.85422 | | | | |
| **Pseudo R²** | | 0.1173 | | | | |

[1]. Reference category.

However, Jacob et al. [1] in India and Mursy and Mohamed [15] in Sudan reported a good level of attitude towards hepatitis B infection prevention. The differences in the findings of the various studies could be related to the nursing students' relative level of perception of risk, severity and threat from hepatitis B infection [9] as well as the respondent's readiness to accept change as advanced by the five-stage cyclical transtheoretical model [35].

The overall practice of hepatitis B infection prevention was poor. This finding confirms those of Afihene et al. [11] in Ghana, Rathi et al. [18] in India and Zaeri et al. [12] in the Kingdom of Saudi Arabia. Poor practice of hepatitis B infection prevention implies a lack of commitment to combating hepatitis B infection. Poor practices increase the nursing student's exposure to the disease and their vulnerability to contracting and/spreading the disease among their potential clients. This, therefore, necessitates advocacy for institution-based policies for regular practical skills training on hepatitis B infection prevention as well as compulsory hepatitis B screening, vaccination and post-vaccination testing among all health trainees. However, Mursy and Mohamed [15] in Sudan and Khan et al. [36] in Bangladesh refute our finding. These differences are plausibly related to the differences in the level of knowledge and attitude among the respondents [8] and also to the social cognitive theory [37]. The respondents intention to perform good practice of hepatitis B infection prevention is dependent on the triadic reciprocal determinism of behaviour, attitude and environment; observational learning; self-efficacy and outcome expectancy [37]. Therefore, management of nursing training colleges should promote education on hepatitis B infection prevention as well as enforce proper professional conduct among the nursing students.

High knowledge of hepatitis B infection prevention was statistically significantly associated with an increased likelihood of exhibiting good practice of hepatitis B infection prevention but only at the 10% significance level. This association has been confirmed by Osei et al. [27] in Ghana, Rathi et al. [18] in India, and Zaeri et al. [12] in the Kingdom of Saudi Arabia. This implies that the acquisition of adequate knowledge plausibly increases health consciousness [38] thereby enabling uptake of hepatitis B infection prevention activities such as avoiding NSI via recapping of needles after use among others. High knowledge of hepatitis B infection prevention enables the nursing students to develop and sustain the requisite skills needed for hepatitis B infection prevention which can empower them to exercise control over their own health [39].

Good attitude towards hepatitis B infection prevention was highly statistically significantly associated with a higher likelihood of demonstrating good practice of hepatitis B infection prevention. This finding is consistent with those of Ul-Haq et al. [19] in Pakistan, Afihene et al. [11] in Ghana and Zaeri et al. [12] in the Kingdom of Saudi Arabia. This means that good attitude towards hepatitis B infection prevention, in the context of perceived susceptibility, perceived severity and perceived threat, as proposed by the health belief model can directly influence the nursing students in respect of the performance of good practice of hepatitis B infection prevention [9]. Additionally, the KAP-O framework proposes that attitude may interact with the knowledge to produce the practice of hepatitis B infection prevention [8]. Therefore, in promoting the health of nursing students, there is the need for trainee nurses to consciously adopt healthy attitudinal postures towards strengthening individual and community action for health promotion.

Being a male was also statistically significantly associated with an increased likelihood of exhibiting good practice of hepatitis B infection prevention. This finding is analogous to that of Ahmad et al. [40] in Malaysia. However, the findings of Abiola et al. [13] in Nigeria and Al-Shamri et al. [41] in the Kingdom of Saudi Arabia, were different. Male nursing students like male nurses remain the minority in the nursing practice [42]. Therefore, male nursing students

in an attempt to remain relevant, are probably compelled to exhibit good practice of hepatitis B infection prevention which culminates in effective hepatitis B infection prevention [8].

Married nursing students were statistically significantly more likely to demonstrate good practice of hepatitis B infection prevention. Analogous finding was reported by Ahmad et al. [40] in Malaysia but the findings of Paudel et al. [43] in Nepal and Abiola et al. [13] in Nigeria were different. Marriage creates a sense of responsibility that compels people to be health conscious [44] and hence to prioritise the promotion of the health and well-being of themselves and their families [40] by taking up disease prevention activities including that of hepatitis B infection prevention. However, we recommend that further studies be conducted to explore the detailed role marriage plays in promoting good practice of hepatitis B infection prevention.

Second year nursing students were also statistically significantly associated with a higher likelihood of exhibiting good practice of hepatitis B infection prevention. This finding is comparable to that of Jacob et al. [1] in India, although it was different from that of Osei et al. [27] in Ghana. This implies that long years of exposure to education and training confers the requisite knowledge and skills needed for adhering to good practice of hepatitis B infection prevention.

Tertiary education status of the students' parent/guardian was also found to be statistically significantly associated with an increased likelihood of performing good practice of hepatitis B infection prevention. This finding supports that of Mora and Trapero [45] in Spain but contrasts the finding of Giao et al. [46] in Vietnam. This finding is explainable in the context of relative access to information and understanding of the importance of hepatitis B infection prevention. Plausibly, highly educated parents or guardians, have greater access and understanding of health issues like hepatitis B infection prevention, from the media including the internet [38]. They are therefore significantly advantaged compared to their counterparts who are unable to access the requisite information. Furthermore, highly educated parents plausibly take the initiative to locate and pay for hepatitis B infection preventive services such as hepatitis B screening, vaccination and post-vaccination services, for their wards.

In line with the findings of this study, several existing studies have also reported no statistically significant association between, age [11], religion [13], programme of study [27] and the practice of hepatitis B infection prevention. However, other studies have reported age [13], urban residential status [19] and name of college [12] to be statistically significantly associated with good practice of hepatitis B infection prevention although such associations were not statistically significant in our study.

## Strengths and limitations of the study

Our KAP survey like all studies has peculiar strengths and weakness. To ensure the internal validity of the results, a standardised questionnaire was validated and tested for internal consistency. Recognising the low response rate associated with online surveys, follow up calls were made to ensure that many of the potential non-responders voluntarily completed the questionnaire [23]. Respondents who had no access to data for internet connectivity were provided hard copies of the questionnaire. Therefore, the response rate of 98.2% obtained in this study, improved the external validity of our study. However, although our study used a multistage stratified random sampling technique, the representativeness of the study cannot be completely vouched for, since only two out of seven colleges of nursing in the Upper West Region of Ghana were sampled. Finally, since our KAP survey relied on self-reported responses, the results of our study may not fully represent reality but the assumption of idealism.

## Conclusion

We conclude that the KAP scores of hepatitis B infection prevention among the nursing students were sub-optimal. High knowledge and good attitude towards hepatitis B infection prevention, moderated by sex, marital status, level of study and parent's level of education, influenced the practice of hepatitis B infection prevention. To effectively prevent hepatitis B infection and promote the health of nursing students, we recommend advocacy and the implementation of institution-based policies for hepatitis B infection prevention; creation of enabling environment through hepatitis B infection prevention education, provision of free or subsidised hepatitis B infection screening, treatment and vaccination services; strengthening of the nursing community action; development of hepatitis B infection prevention skills such as prevention of needle stick injuries and the enforcement of proper professional ethics. These disease prevention interventions should be streamlined towards the specific needs of female nursing students, nursing students who are not married, first year nursing students and those whose parents/guardians' educational levels are below tertiary.

## Supporting information

**S1 Table. KAP scoring sheet.**
(PDF)

**S1 Questionnaire. KAP of hepatitis B infection prevention survey.**
(PDF)

**S1 Data. Dataset from pre-testing.**
(DTA)

**S2 Data. Minimum dataset from survey.**
(XLS)

## Acknowledgments

We acknowledge the immense support of Linda Bayaa, Seidu Zakaria, Shiela Konadu, Richard Basadi, Ahmadu Zakaria, Paschal Kob, Albert Buondau, Albin Daabadi, Doreen Anzunna and Getrude Tempare, during the data collection.

## Author Contributions

**Conceptualization:** Augustine Ngmenemandel Balegha, Gilbert Abotisem Abiiro.

**Data curation:** Augustine Ngmenemandel Balegha, Gilbert Abotisem Abiiro.

**Formal analysis:** Augustine Ngmenemandel Balegha.

**Investigation:** Augustine Ngmenemandel Balegha, Adadow Yidana.

**Methodology:** Augustine Ngmenemandel Balegha.

**Project administration:** Augustine Ngmenemandel Balegha.

**Resources:** Augustine Ngmenemandel Balegha.

**Software:** Augustine Ngmenemandel Balegha, Gilbert Abotisem Abiiro.

**Supervision:** Augustine Ngmenemandel Balegha, Adadow Yidana, Gilbert Abotisem Abiiro.

**Validation:** Augustine Ngmenemandel Balegha, Adadow Yidana, Gilbert Abotisem Abiiro.

**Visualization:** Augustine Ngmenemandel Balegha, Adadow Yidana, Gilbert Abotisem Abiiro.

**Writing – original draft:** Augustine Ngmenemandel Balegha.

**Writing – review & editing:** Augustine Ngmenemandel Balegha, Adadow Yidana, Gilbert Abotisem Abiiro.

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
