## [Decision Letter · Decision Letter 0]

27 Jul 2021

PONE-D-21-21003

KNOWLEDGE, ATTITUDE AND PRACTICE OF HEPATITIS B INFECTION PREVENTION AMONG NURSING STUDENTS IN GHANA: A CROSS SECTIONAL STUDY

PLOS ONE

Dear Dr. Belegha,

Thank you for submitting your manuscript to PLOS ONE. After careful consideration, we feel that it has merit but does not fully meet PLOS ONE’s publication criteria as it currently stands. Therefore, we invite you to submit a revised version of the manuscript that addresses the points raised during the review process.

Many thanks for submitting your manuscript to PLOS One

It was reviewed by three experts in the field, and they have recommended some modifications be made prior to acceptance

I therefore invite you to make these changes and to write a response to reviewers which will expedite revision upon resubmission

I wish you the best of luck with your modifications

Hope you are keeping safe and well in these difficult times

Thanks

Simon

We look forward to receiving your revised manuscript.

Kind regards,

Simon Clegg, PhD

Academic Editor

PLOS ONE

2. Please include additional information regarding the survey or questionnaire used in the study and ensure that you have provided sufficient details that others could replicate the analyses. For instance, if you developed a questionnaire as part of this study and it is not under a copyright more restrictive than CC-BY, please include a copy, in both the original language and English, as Supporting Information. If the original language is written in non-Latin characters, for example Amharic, Chinese, or Korean, please use a file format that ensures these characters are visible.

3. Please state whether you validated the questionnaire prior to testing on study participants. Please provide details regarding the validation group within the methods section.

Reviewers' comments:

Reviewer's Responses to Questions

**Comments to the Author**

1. Is the manuscript technically sound, and do the data support the conclusions?

Reviewer #1: No

Reviewer #2: Yes

Reviewer #3: Yes

2. Has the statistical analysis been performed appropriately and rigorously? 

Reviewer #1: No

Reviewer #2: No

Reviewer #3: Yes

3. Have the authors made all data underlying the findings in their manuscript fully available?

Reviewer #1: Yes

Reviewer #2: Yes

Reviewer #3: Yes

4. Is the manuscript presented in an intelligible fashion and written in standard English?

Reviewer #1: Yes

Reviewer #2: No

Reviewer #3: Yes

5. Review Comments to the Author

Reviewer #1: KNOWLEDGE, ATTITUDE AND PRACTICE OF HEPATITIS B INFECTION PREVENTION AMONG NURSING STUDENTS IN GHANA: A CROSS SECTIONAL STUDY

The analysis performed for regression is very vague and the results are not consistent with the objectives. It is also difficult for the reader to understand (Table 6). Multivariate logistic regression was not performed. The model assumptions in the materials and methods should be written in full and it should be noted how this analysis was performed (more details).

Is there a significant difference in the study of people (demographic variables) in age group, gender, marital status and place of residence, etc. (Table 1)? Because these variables are confusing, their differences can affect multivariate regression analysis.

Reviewer #2: COMMENTS TO THE AUTHOR(S):

Overall:

1. It is advised to take the assistance of native speakers or editing services to improve the quality of writing. I could see many fundamental write-up-related issues like correct grammar and tense, spell out of numbers below ten, complete form in the first use of abbreviations, and inappropriate syntax, etc. Even if authors can’t afford the editing services, they are advised to use Grammarly premium to omit such basics. Additionally, write is up is too much lengthy. The introduction must be concise, justifying the study, and usually of three average paragraphs. But, the study is almost three pages. So, please omit too much unimportant information (waste words) and findings of other studies in detail. However, detailing the method is expected and justified!

2. Rest as listed

Specific:

1. Topic: The study participants are being taken from only two colleges from a specific region of the country. So, generalization for the whole nation may not be appropriate and relevant. Therefore, it would be better not to use the term “in Ghana” in the topic.

2. Introduction: It is too lengthy to go through. Even though there are no word constraints in journal guidelines, it is better to concise and condense the introduction. Just use the required information only.

3. Results:

There is a difference in the percentage of ‘never married’ participants in tables and descriptions in words.

It is described that “…the nursing students recognized MTCT (79.8%), unprotected sex (81.1%) and unsterilized surgical instruments (85.6%) as the major route of transmission of hepatitis B infection.” And, what about contaminated blood/products? Again, 90.8% have recognized it as the significant route!!

The term “48.5% knew hepatitis B vaccine as an anticancer vaccine” sounds unusual. So you had better write 48.5% of them knew the hepatitis B vaccine protects against liver cancer.

While describing any finding, it is better to use the parameter with a higher value first, followed by lower values.

4. Statistics:

In table 5, both mean (SD) and median (IQR) for composite scores are kept without any statistical consideration. Generally, we use mean and standard deviation for normal data, and for non-normally distributed data, we use the median (IQR) for continuous variables. Would you please mention clearly in method as well regarding distribution of data and appropriate use of statistical tests?

5. Discussion:

Comparison and contrast with other studies have been made extensively. Therefore, please add “take away message” from your research to the existing medical literature.

Reviewer #3: It is well written article, with a very good sample size.

Here are my comments, hope these will help in improvement of manuscript and quality of information.

Abstract: Overall abstract do not represent the study presented but need correction. Editing of English is advised as using a passive voice instead of active voice.

Background: The abstract mentioned the background as “Good knowledge, attitude and practice (KAP) of hepatitis B infection (HBI) prevention are essential to disease prevention" which is not sufficient as a background of the study.

Methodology: Methodology is not clearly presented in abstract, need correction.

Reference: Authors are advised to check the reference for completeness and journal requirements.

6. PLOS authors have the option to publish the peer review history of their article (what does this mean?). If published, this will include your full peer review and any attached files.

Reviewer #1: No

Reviewer #2: No

---

## [Author Response · Author response to Decision Letter 0]

28 Aug 2021

POINT-BY-POINT RESPONSE TO REVIEWERS’ COMMENTS

Ref: Submission ID [PONE-D-21-21003] - [EMID:4da1bd33a6fc8347]

Title: Knowledge, attitude and practice of hepatitis B infection prevention among nursing students in the Upper West Region of Ghana: a cross-sectional study

Editorial comments

General response

We are very grateful to the editor for the positive feedback regarding our manuscript. We are also thankful for the editorial comments which we have addressed. We hope that our manuscript now meets the editorial requirements of the journal.

Comment 1

Please ensure that your manuscript meets PLOS ONE’s style requirements, including those for file naming.

Response 1

Thank you for this reminder. We have referred to PLOS One’s style requirements as exemplified by the PLOS One templates for title page and manuscript body formatting guidelines. We have also named our files according to the requirements of the journal.

Comment 2

Please include additional information regarding the survey or questionnaire used in the study and ensure that you have provided sufficient details that others could replicate the analyses.

Response 2 

We are grateful for this guideline. We have included more information on the survey and the questionnaire as found under “data collection” of the methods section. We have also added the questionnaire as a supplementary information file.

Comment 3

Please state whether you validated the questionnaire prior to testing on study participants. Please provide details regarding the validation group within the methods section. 

Response 3

Thank you for this comment. The questionnaire was validated before data collection. Details of the validation group and the validation process can now be found in the methods section under a fresh sub-heading “validity and reliability of the instrument”.

Comment 4

In your Data Availability statement, you have not specified where the minimum data set underlying the results described in your manuscript can be found. Upon re-submitting your revised manuscript, please upload your study’s minimal underlying data set as either Supporting Information files or to a stable, public repository and include the relevant URLs, DOIs, or accession numbers within your revised cover letter. We will update your Data Availability statement to reflect the information you provide in your cover letter. 

Response 4

We are grateful for your guidelines. We have now uploaded the minimum underlying data set of our study as a supporting information file.

Comment 5

Your ethics statement should only appear in the Methods section of your manuscript. If your ethics statement is written in any section besides the methods, please move it to the methods section and delete it from any other section. 

Response 5

As required, we have moved the ethics statement of this study to the methods section. Consequently, we have deleted it from its previous position on the original manuscript.

Reviewers’ comments

Reviewer 1

Comment 1

The analysis performed for regression is very vague and the results are not consistent with the objectives. It is also difficult for the reader to understand (Table 6).

Response 1

We thank the reviewer for the comment. We have revised the wording of some parts of the results to improve upon clarity of the interpretation of the results from the regression model presented in Table 6. We think that the regression analysis we performed is not inconsistent with our study objectives. The last objective of our study was to assess the factors associated with the practice of hepatitis B infection prevention. The dependent variable “practice of Hepatitis B infection prevention” had three main outcomes which were ordered in nature. The multivariate generalised logistic regression analysis we performed served to assess the association between practice of hepatitis B infection prevention as the dependent variable, and knowledge of hepatitis B infection prevention, attitude towards hepatitis B infection prevention and the socio-demographic characteristics of the respondents as the independent variables

Table 6 displays the results of the multivariate generalised ordered logistic regression model we run. Although the result was from the running of a single model, it generated two outputs in accordance with the partial parallel regression/ partial proportional odds assumption. We have clearly explained this under the data analysis part of the methods section.

To further improve upon the clarity of the results in Table 6, we have revised our interpretations of the results in the main manuscript.

Comment 2 

Multivariate logistic regression was not performed.

Response 2

We thank the reviewer for this comment. Perhaps our wording of some parts of the methods section that was not clear to the reviewer but we indeed performed multivariate logistic regression. Based on our understanding of multivariate logistic regression, we think that the generalised ordered logistic regression modelling that we performed is a form of multivariate logistic regression. As indicated above, our dependent variable, practice of hepatitis B infection prevention is a polychotomous ordered categorical variable (poor, moderate and good practice of hepatitis B infection prevention). Therefore, initially a multivariate ordered logistic regression was performed by regressing practice of hepatitis B infection on knowledge of hepatitis B infection, attitude towards hepatitis B infection and the socio-demographic characteristics of the respondents. As explained under data analysis of methods section, the multivariate ordered logistic regression model failed the Brant test of proportional odds assumption. A multivariate generalised ordered logistic regression model with autofit option was the best alternative as cited by the relevant literature under methods section. In sum, the regression modelling that we performed was multivariate.

Comment 3

The model assumptions in the materials and methods section should be written in full and it should be noted how this analysis was performed (more details). 

Response 3

Thank you for your comment. As rightly pointed out by the reviewer, we have clearly and fully explained the parallel regression/ proportional assumption as well as the partial proportional odds assumptions under data analysis of the methods section. We have also indicated in detail under data analysis, how the analysis was performed. In fact, the entire analysis subsection of the methods has been rewritten to improve upon clarity of our methodological descriptions.

Comment 4 

Is there a significant difference in the study of people (demographic variables) in age group, gender, marital status and place of residence, etc. (Table 1)? Because these variables are confusing, their differences can affect multivariate regression analysis.

Response 4

We are grateful for your comment. Table 1 displays a descriptive analysis of the socio-demographic characteristics of the respondents in frequencies and percentages. The analysis presented in Table 1 was univariate for the purpose of only describing the proportions of respondents under each category of each variable. We did not conduct bivariate analysis to show differences between socio-demographic variables and any dependent variable (i.e practice of hepatitis B infection). We explored such significant associations (differences) in the multivariate regression modelling. As indicated by Table 6, the socio-demographic characteristics of the respondents which showed statistically significant associations when regressed on practice of hepatitis B infection prevention in our multivariate generalised logistic regression model were; sex, marital status, level of education and parent’s level of education. We described these significant associations of the socio-demographic characteristics with the practice of hepatitis B in the results section and further discussed these findings under the discussion section. 

Reviewer 2

Comment 1 

It is advised to take the assistance of native speakers or editing services to improve the quality of writing. I could see many fundamental write-up-related issues like correct grammar and tense, spell out of numbers below ten, complete form in the first use of abbreviations, and inappropriate syntax, etc. Even if authors can’t afford the editing services, they are advised to use Grammarly premium to omit such basics. 

Response 1

We thank the reviewer these valuable comments. We have edited and proof read our manuscript with Grammarly premium.

Comment 2 

Additionally, write is up is too much lengthy. The introduction must be concise, justifying the study, and usually of three average paragraphs. But, the study is almost three pages. So, please omit too much unimportant information (waste words) and findings of other studies in detail. However, detailing the method is expected and justified! 

Response 2

Thank you for your comment. We agree with you that the write-up, especially the introduction was lengthy. We have therefore identified and deleted some irrelevant statements under the introduction section.

Comment 3 

Topic: The study participants are being taken from only two colleges from a specific region of the country. So, generalization for the whole nation may not be appropriate and relevant. Therefore, it would be better not to use the term “in Ghana” in the topic. 

Response 3

We agree with the reviewer’s comment. We have rephrased the title as “Knowledge, attitude and practice of hepatitis B infection prevention among nursing students in the Upper West Region of Ghana: a cross-sectional study. We have specified the region from which the study was conducted.

Comment 4

Introduction: It is too lengthy to go through. Even though there are no word constraints in journal guidelines, it is better to concise and condense the introduction. Just use the required information only.

Response 4

We agree with the reviewer that the write-up, especially the introduction section was lengthy. We have therefore identified and deleted some less important statements under the introduction section.

Comment 5 

Results:

There is a difference in the percentage of ‘never married’ participants in tables and descriptions in words.

Response 5

Thank you for your comment. We have corrected the error in reporting the proportion of “never married” respondents. The correct figure is 90.8 %.

Comment 6

It is described that “…the nursing students recognized MTCT (79.8%), unprotected sex (81.1%) and unsterilized surgical instruments (85.6%) as the major route of transmission of hepatitis B infection.” And, what about contaminated blood/products? Again, 90.8% have recognized it as the significant route!!

Response 6

We have now reported the proportion of respondents who recognised contaminated blood/ products as a transmission route for hepatitis B infection as 90.8%.

Comment 7 

The term “48.5% knew hepatitis B vaccine as an anticancer vaccine” sounds unusual. So you had better write 48.5% of them knew the hepatitis B vaccine protects against liver cancer.

Response 7

We are grateful for this comment. We have re-written the statement to read “48.5% of them knew that hepatitis B vaccine protects against liver cancer”.

Comment 8 

While describing any finding, it is better to use the parameter with a higher value first, followed by lower values. 

Response 8

We have now reported higher values of parameters of the results before the lower values.

Comment 9 

Statistics:

In table 5, both mean (SD) and median (IQR) for composite scores are kept without any statistical consideration. Generally, we use mean and standard deviation for normal data, and for non-normally distributed data, we use the median (IQR) for continuous variables. Would you please mention clearly in method as well regarding distribution of data and appropriate use of statistical tests?

Response 9

We are grateful for comment. The composite scores for knowledge, attitude and practice of hepatitis B infection prevention were not normally distributed. Therefore, although several summary statistics have been reported for KAP of hepatitis B infection prevention, composite median scores with interquartile ranges (IQRs) have now been used as the measure of central tendency for assessing KAP of hepatitis B infection prevention. We have therefore replaced the mean and standard deviation values with the median and IQR values in the abstract and in the description of the results in the main manuscript. Additionally, the distribution as well as appropriate statistical tests of the data for KAP of hepatitis B infection prevention have now been stated clearly under data analysis of the methods section. Consequently, we have included skewness and kurtosis as measures of data distribution under both the methods and results sections.

Comment 10 

Discussion:

Comparison and contrast with other studies have been made extensively. Therefore, please add “take away message” from your research to the existing medical literature. 

Response 10

We thank reviewer for the comment. We think that we have already presented “take away messages in the form of practical policy and research implications in the discussions of the results. The last part of each paragraph in the discussion section sought to bring out the implications of our findings in the form of take-away messages. However, in addition to what we have already presented we have included “Therefore, management of nursing training colleges should promote education on hepatitis B infection prevention as well as enforce proper professional conduct among the nursing students” as a policy implication under the discussion for the overall poor practice of the nursing students. We have also included “However, we propose that further studies be conducted to explore the detailed role marriage plays in promoting good practice of hepatitis B infection prevention” as an implication for further studies.

Reviewer 3

Comment 1 

It is well written article, with a very good sample size. Here are my comments, hope these will help in improvement of manuscript and quality of information.

Response 1

We are very grateful to the reviewer for appreciating our work. We are also thankful for your constructive comments offered to help improve the quality of our work.

Comment 2 

Abstract: Overall abstract do not represent the study presented but need correction. Editing of English is advised as using a passive voice instead of active voice.

Response 2

Thank you for your comment. We have revised the abstract as a whole especially the background, results and conclusion of the abstract. To the best of our knowledge, we have now used active voice in our write-up.

Comment 3 

Background: The abstract mentioned the background as “Good knowledge, attitude and practice (KAP) of hepatitis B infection (HBI) prevention are essential to disease prevention" which is not sufficient as a background of the study.

Response 3

We have added that “Hepatitis B infection remains a public health threat associated with undesirable statistics of morbidity and mortality”. Therefore, the background of the abstract now reads “Hepatitis B infection remains a public health threat associated with undesirable statistics of morbidity and mortality. Good knowledge, attitude and practice (KAP) of hepatitis B infection (HBI) prevention are essential to hepatitis B infection control. However, there is limited evidence on the KAP of HBI prevention among nursing students, who are significantly exposed to HBI. We assessed KAP of HBI prevention and the factors associated with the practice of HBI prevention”.

Comment 4 

Methodology: Methodology is not clearly presented in abstract, need correction. 

Response 4

We are grateful for the comment. The methods section of our manuscript covered the study setting, design, population and sampling, data collection method and period and data analysis including software and version used for data analysis of both the descriptive and inferential statistical analysis performed. Therefore, to the best of our knowledge, we think that the methods section of the abstract exhausted the relevant aspects of a typical methods section of an abstract.

Comment 5 

Reference: Authors are advised to check the reference for completeness and journal requirements, 

Response 5

We are grateful for your constructive comment. We have inspected the references and ensured their completeness and conformation with the referencing style recommended by PLOS ONE.

---

## [Decision Letter · Decision Letter 1]

21 Sep 2021

PONE-D-21-21003R1Knowledge, attitude and practice of hepatitis B infection prevention among nursing students in the Upper West Region of Ghana: a cross-sectional studyPLOS ONE

Dear Dr. Belegha,

Thank you for submitting your manuscript to PLOS ONE. After careful consideration, we feel that it has merit but does not fully meet PLOS ONE’s publication criteria as it currently stands. Therefore, we invite you to submit a revised version of the manuscript that addresses the points raised during the review process.

Many thanks for submitting your manuscript to PLOS One

It was reviewed by two experts in the field, and they have recommended some modifications be made prior to acceptance

I therefore invite you to make these changes and to write a response to reviewers which will expedite revision upon resubmission

I wish you the best of luck with your modifications

Hope you are keeping safe and well in these difficult times

Thanks

Simon

We look forward to receiving your revised manuscript.

Kind regards,

Simon Clegg, PhD

Academic Editor

PLOS ONE

Journal Requirements:

Reviewers' comments:

Reviewer's Responses to Questions

**Comments to the Author**

1. If the authors have adequately addressed your comments raised in a previous round of review and you feel that this manuscript is now acceptable for publication, you may indicate that here to bypass the “Comments to the Author” section, enter your conflict of interest statement in the “Confidential to Editor” section, and submit your "Accept" recommendation.

Reviewer #1: All comments have been addressed

Reviewer #2: All comments have been addressed

2. Is the manuscript technically sound, and do the data support the conclusions?

Reviewer #1: No

Reviewer #2: Yes

3. Has the statistical analysis been performed appropriately and rigorously? 

Reviewer #1: No

Reviewer #2: Yes

4. Have the authors made all data underlying the findings in their manuscript fully available?

Reviewer #1: Yes

Reviewer #2: Yes

5. Is the manuscript presented in an intelligible fashion and written in standard English?

Reviewer #1: Yes

Reviewer #2: Yes

6. Review Comments to the Author

Reviewer #1: Please have the data analyzed by a statistician or epidemiologist and the results reviewed. There is still ambiguity in data analysis. Materials and methods are also confusing.

Reviewer #2: Authors have rectified the comments and suggestions in their previous version of the manuscript, so the current version of the manuscript can be considered for the further processing.

7. PLOS authors have the option to publish the peer review history of their article (what does this mean?). If published, this will include your full peer review and any attached files.

Reviewer #1: No

Reviewer #2: No

---

## [Author Response · Author response to Decision Letter 1]

28 Sep 2021

POINT-BY-POINT RESPONSE TO REVIEWERS’ COMMENTS

Ref: Submission ID [PONE-D-21-21003R1] - [EMID:72516b38fa838aa1]

Title: Knowledge, attitude and practice of hepatitis B infection prevention among nursing students in the Upper West Region of Ghana: a cross-sectional study

Editorial comments

General response

We are very grateful to the editor for the positive feedback regarding our manuscript. We hope that this current version of our manuscript will be suitable for publication.

Journal Requirements:

Response

We have reviewed and updated our reference list. The following are the modifications made;

Reference 1

The page numbers of the first reference has been updated to “xxx-xxx” in the current version of the manuscript. The first reference now reads; 

1. Jacob A, Joy TM, Mohandas S, Lais H, Paul N. Assessment of knowledge, attitude, and practice regarding hepatitis B among medical students in a private medical college in Kochi. Int J Community Med Public Health. 2019;6: xxx–xxx. doi:10.18203/2394-6040.ijcmph20191552

Reference number 4

We have updated the list of contributors of the 4th reference in the previous manuscript. We have included “Apica B, Awuku Y, Cunha L, et al.” The reference now reads; Spearman CW, Afihene M, Ally R, Apica B, Awuku Y, Cunha L, et al. Hepatitis B in sub-Saharan Africa: strategies to achieve the 2030 elimination targets. The Lancet Gastroenterology & Hepatology. 2017;2: 900–909. doi:10.1016/S2468-1253(17)30295-9

Reference number 6

“Doi” in reference number six was duplicated in the previous version of the manuscript as “doi.doi”. Therefore, we have reviewed it to read; Dongdem JT, Kampo S, Soyiri IN, Asebga PN, Ziem JB, Sagoe K. Prevalence of hepatitis B virus infection among blood donors at the Tamale Teaching Hospital, Ghana (2009). BMC Res Notes. 2012;5: 115. doi:10.1186/1756-0500-5-115

Reference number 21

We have deleted “1” depicted wrongly as the page number of the 21st reference in the previous version of the manuscript. The reference now reads; Al-Hussami M. Knowledge and acceptance of hepatitis B vaccine. Internet Journal of Health Care Administration. 2001;2. Available: https://ispub.com/IJHCA/2/1/5725

Reference numbers 23, 24 and 25

Reference numbers 23, 24 and 25 in the previous manuscript have been correctly ordered in the newly revised manuscript. These references are arranged in the previous manuscript as follows;

23. Daud KAM, Khidzir NZ, Ismail AR, Abdullah FA. Validity and reliability of instrument to measure social media skills among small and medium entrepreneurs at Pengkalan Datu River. 2018;7: 1026–1037. 

24. Cronbach LT. Coefficient alpha and the internal structure of tests. Psychometrika. 1951;16: 297–334. doi:10.1007/BF02310555

25. Creswell JW, Creswell JD. Research Design: Qualitative, Quantitative, and Mixed Methods Approach. 5th ed. Los Angeles: Sage Publications, Inc.; 2018. 

However, after our update the references were re-organised as follows;

23. Creswell JW, Creswell JD. Research Design: Qualitative, Quantitative, and Mixed Methods Approach. 5th ed. Los Angeles: Sage Publications, Inc.; 2018. 

24. Daud KAM, Khidzir NZ, Ismail AR, Abdullah FA. Validity and reliability of instrument to measure social media skills among small and medium entrepreneurs at Pengkalan Datu River. 2018;7: 1026–1037. 

25. Cronbach LT. Coefficient alpha and the internal structure of tests. Psychometrika. 1951;16: 297–334. doi:10.1007/BF02310555

Reference number 28

The page numbers of reference number 28 which was written as 1171-78 has now been fully written. It therefore now reads; 

Brant R. Assessing proportionality in the proportional odds model for ordinal logistic regression. Biometrics. 1990;46: 1171–1178. doi:10.2307/2532457

Reference numbered 31

After our review, we realised that the 31st reference on the previous version of the manuscript had previously been deleted and therefore not in the manuscript text. Consequently, we have deleted it from the current version. However, the remaining references have been reorganised as their positions have been changed due to the deletion.

Reference 36

Reference 36 in the previous version of the manuscript has been updated to a later edition of the book chapter. Therefore, the 36th reference previously listed as

“Prochaska JO, Redding C, Evers K. The transtheoretical model and stage of change. Health behavior and health education Theory, Research, and Practice. 2008. pp. 99–120. Available: https://soh.iums.ac.ir/uploads/beeduhe_0787996149.pdf#page=135” has been updated to; 

Prochaska JO, Redding C, Evers K. The transtheoretical model and stages of change. 5th ed. In: K Glanz, B K Rimer, & K “V” Viswanath, editors Health behavior: Theory, research, and practice. 5th ed. Jossey-Bass/Wiley; 2015. pp. 125–148.”

Reference 38

We added “In:” to the 38th reference on the previous version of the manuscript, now 37th reference in the revised manuscript. We also replaced “(Eds)” with “editors”. The page number were also written in full as “165-184” and not “165-84”. The reference now reads;

Baranowski T, Perry CL, Parcel GS. How individuals, environments, and health behavior interact: Social cognitive theory. 3rd ed. In: K Glanz, BK, Rimer & FM Lewis, editors Health Behavior and health education: Theory, research, and practice. 3rd ed. San Francisco: Jossey-Bass; 2002. pp. 165–184.

Reference number 40

Reference number 40 of the previous version of the manuscript, now reference number 39 of the current version, has been updated from

“Naidoo J, Wills J. Foundations of Health Promotion. Oxford: Baillière Tindall; 2009. Available: http://pdfpremiumfree.com/download/naidoo-and-wills-2009-foundations-of-health-promotion-pdf-pdf/” to; 

“Naidoo J, Wills J. Health Promotion: Foundations for Practice. 3rd ed. London: Baillière Tindall; 2009.”

Reference number 44

The page numbers of reference number 44 of the previous version of the manuscript, now number 43 on the current version, has been written in full as “109-113” instead of “109-13”. The reference now reads;

Paudel D, Prajapati S, Paneru D. Preventive practices against Hepatitis B: A cross-sectional study among nursing students of Kathmandu, Nepal. J Sci Soc. 2012;39: 109–113. doi:10.4103/0974-5009.105911

Reviewers' comments:

General response

We are very grateful to the reviewers for the positive feedback concerning our manuscript. We are particularly very excited that both reviewers have acknowledged that we have addressed all comments. 

Reviewer #1: 

Please have the data analyzed by a statistician or epidemiologist and the results reviewed. There is still ambiguity in data analysis. Materials and methods are also confusing.

Response

We thank the reviewer for acknowledging that we have addressed all comments raised in our previous submission. Although, the reviewer suggested that our data be re-analysed by a statistician or epidemiologist, no specific issues were raised.

Nevertheless, the authors have background knowledge in statistics and epidemiology as well as regression modelling. We have already cited William (2006 & 2016), an authority in the analysis and interpretation of data involving ordered categorical outcome variables. Using STATA, William (2006 & 2016) explained the methods for such analysis and the interpretation of the data output. We are therefore convinced that our analysis and interpretation of results is appropriate. 

The methods and material section covers the KAP-O framework underpinning the study, the study setting, approach and design, sampling, data collection, validity and reliability of study instrument, ethical considerations as well as data analysis. We believe therefore that we have adequately explained the methods we used in our study. We also agree with the unanimous comment by both reviewers that our manuscript has been presented in an intelligible fashion and written in standard English. Therefore, we believe that the methods section as well as the entire manuscript is intelligible enough to be comprehended by the reader.

Reviewer #2

Authors have rectified the comments and suggestions in their previous version of the manuscript, so the current version of the manuscript can be considered for the further processing.

Response

We are very grateful to the reviewer for this comment. We appreciate the reviewer for commending our work. We are even much happier for the recommendation that our manuscript be considered for further processing.

---

## [Editor Report · Decision Letter 2]

5 Oct 2021

Knowledge, attitude and practice of hepatitis B infection prevention among nursing students in the Upper West Region of Ghana: a cross-sectional study

PONE-D-21-21003R2

Dear Dr. Belegha,

We’re pleased to inform you that your manuscript has been judged scientifically suitable for publication and will be formally accepted for publication once it meets all outstanding technical requirements.

Kind regards,

Simon Clegg, PhD

Academic Editor

PLOS ONE

Additional Editor Comments:

Many thanks for resubmitting your manuscript to PLOS One

As you have addressed all the comments and the manuscript reads well, I have recommended it for publication

You should hear from the Editorial Office shortly.

It was a pleasure working with you and I wish you the best of luck for your future research

Hope you are keeping safe and well in these difficult times

Thanks

Simon

---

## [Editor Report · Acceptance letter]

6 Oct 2021

PONE-D-21-21003R2 

Knowledge, attitude and practice of hepatitis B infection prevention among nursing students in the Upper West Region of Ghana: a cross-sectional study 

Dear Dr. Balegha:

I'm pleased to inform you that your manuscript has been deemed suitable for publication in PLOS ONE. Congratulations! Your manuscript is now with our production department. 

Kind regards, 

on behalf of

Dr. Simon Clegg 

Academic Editor

PLOS ONE